# Isolation and Identification of *Aeromonas veronii* in Sheep with Fatal Infection in China: A Case Report

**DOI:** 10.3390/microorganisms11020333

**Published:** 2023-01-29

**Authors:** Yongqiang Miao, Xueliang Zhao, Fathalrhman Eisa Addoma Adam, Qingfang Xie, Hang Feng, Jingru Ding, Xindong Bai, Juan Wang, Zengqi Yang

**Affiliations:** College of Veterinary Medicine, Northwest A&F University, Yangling, Xianyang 712100, China

**Keywords:** *Aeromonas veronii*, AMR, respiratory disease, sheep

## Abstract

According to the findings of a sheep breeding farm in Shaanxi, China, 2.53% (15/594) of sheep exhibited respiratory (clinical) symptoms such as dyspnoea, nasal discharge, wet cough, fever, and progressive emaciation. Although multi-drug treatment strategies (including ampicillin, tylosin, florfenicol, and ceftiofur) have been attempted to improve clinical outcomes, they have only been met with limited success, with a mortality rate of 40%. Ultimately, *Aeromonas veronii* (*A. veronii*) was identified as the causative pathogen for respiratory disease. The rates of symptomatic and asymptomatic sheep positive to *A. veronii* were 64.28% (95% CI 52.25–76.31%) and 8.02% (95% CI 6.96–9.08%), respectively. Pathogenicity tests demonstrated that the *A. veronii* is pathogenic to sheep and mice. The results of the antibiotic susceptibility tests revealed that the strain was sensitive to cefotaxime, gentamicin, and enrofloxacin and resistant to ampicillin, ceftiofur, amoxicillin, kanamycin, neomycin, streptomycin, tetracycline, florfenicol, and tylosin. We suggest that the combination of cefotaxime and gentamicin is an effective treatment based on the results of an antimicrobial susceptibility test, which exhibited good therapeutic efficacy. To the best of our knowledge, this is the first report in which pathogenic *A. veronii* has been documented as the cause of death in sheep in China. We concluded that pathogenic *A. veronii* poses a potential risk to the industry of sheep husbandry. This study’s findings can help guide prevention and treatment plans for *A. veronii* infection in sheep.

## 1. Introduction

*A. veronii*, an emerging opportunistic pathogen, is widely present in natural environments, predominantly in freshwater and estuaries [1]. It has been reported that *A. veronii* causes diarrhea, wound infections, pneumonia, peritonitis, uremic syndrome, and hemorrhagic septicemia in humans [2,3,4]. Furthermore, *A. veronii* causes mass mortalities in Nile tilapia and Sea Bass and catastrophic losses to the fish farm industry [5,6]. *A. veronii* infections have also caused the deaths of ducks and foxes [7,8]. However, there is no documentation of clinical symptoms and pathological changes in *A. veronii* causing disease in sheep in China. 

Respiratory diseases lead to high economic losses in the sheep industry worldwide. Several pathogens cause respiratory disease, such as *Pasteurella multocida*, *Mycoplasmosis*, *Mannheimia haemolytica,* and *Staphylococcus aureus* [9,10,11]. In this report, *A. veronii* was first isolated in a sheep that died from a respiratory disease. This intensive sheep farm did not have any sheep trade in recent months. Because of our findings, the list of the pathogenic bacteria that cause respiratory diseases in sheep has been enriched.

## 2. Case Description

One intensive commercial Hu sheep farm in north Shaanxi Province, China, with an altitude range of 1266 m, a longitude range of 109.02°, and a latitude range of 38.20°, reported that about 2.53% (15/594) of the sheep experienced respiratory (clinical) symptoms such as dyspnoea, runny nose, wet cough, and hyperthermia (the average body temperature of the diseased was 41.2 °C), with a mortality rate higher than 40% in the diseased animals, shown in Appendix A. The herd had been vaccinated with the Foot and Mouth Disease Vaccine (Xinjiang TianKang Bio-technique Co., Ltd., Urumqi, China, strain OHM/02 and AKT-III, inactivated) and the Peste des Petits Ruminants Vaccine (Xinjiang TianKang Bio-technique Co., Ltd. strain Clone9, inactivated). Multi-drug treatment strategies (such as penicillin 20,000 U/Kg, ceftiofur 5 mg/kg, and tylosin 10 mg/kg, repeated twice per day for 7 days) have been attempted to improve clinical outcomes, but with limited success.

On 15 April 2022, we went to the farm and happened to have a 15-month-old mature ewe die from respiratory disease. The sheep had been treated with ceftiofur sodium (5 mg/kg body weight, intramuscular, Qilu Animal Health Products Co., Ltd., Jinan, China) for five consecutive days. After the necropsy, we examined all of the organs and found hemorrhage in multiple organs, such as the congestion of the conjunctiva and nasal discharge shown in Figure 1A, a congestive tracheal ring in Figure 1B, the pulmonary consolidation and congestion in Figure 1C, and intestinal mucosal hemorrhage in Figure 1D. In addition, 14 and 187 blood samples were collected and centrifuged (5000× *g*, 10 min) to gather serum in symptomatic and asymptomatic sheep, respectively [12]. The lung from the dead sheep and serum samples were sent to our laboratory under cold-chain standards.

The lung tissues were fixed in a 4% formaldehyde solution. Tissue paraffin embedding, sectioning, and H&E staining were performed by Y&KBio Co., Ltd., Xi’an, China. The sections were observed with an Olympus BX51 microscope (Olympus Corporation, Tokyo, Japan) [13], shown in Figure 2. The pathological findings included infiltrating lymphocytes, fibrinous exudation, thickened alveolar space, and interstitial lung congestion in different parts of the lung from the dead sheep.

A lung sample was used for testing for common respiratory pathogens such as Mycoplasma [14], peste des petits ruminant’s virus (PPRV) [15], foot and mouth disease virus (FMDV) [15], Maedi-Visna (MV) [16], parainfluenza virus type 3 (PIV3) [17], and ovine respiratory syncytial virus (ORSV) [18] using previously reported methods (primer sequences are available in Appendix A), and the results were negative. At the same time, small pieces of lungs were used for pathogenic bacterial isolation. Specifically, inoculation on nutrient agar containing 5% sheep blood was performed using a sterilized inoculation loop dipped in the fresh-cut surface of the lung. After anaerobic and aerobic incubation at 37 °C for 24 h, single-strain isolates were subjected to three rounds of single colony purification. A bacterium strain was isolated in the lung, which is a gram-negative, rod-shaped strain with β-hemolytic, shown in Figure 3.

The 16S rRNA gene of the isolated strain was amplified by the universal primers, 27F:5′-AGAGTTTGATCCTGGCTCAG-3′ and 1492R: 5′-GGTTACCTTGTTACGACTT-3′. The PCR conditions consisted of 94 °C for 5 min, followed by 35 cycles of 94 °C for 30 s, 55 °C for 45 s, 72 °C for 90 s, and a final extension at 72 °C for 6 min. The obtained sequences were sequenced at Beijing Tsingke Biotechnology Co., Ltd, Xi’an, China. A BLAST search for sequences was carried out via the NCBI public database (BLAST: Basic Local Alignment Search Tool). The sequence showed 100% identity (Query cover 98%) with the strain isolated from humans (NR112838) and the standard strain ATCC35624. Moreover, the sequence shares 99.84% identity with the strain isolated from the environment (NR044845). The sequence was submitted to the database at NCBI under accession number ON442310.1. MEGA X software was used to align the sequences and generate the phylogenetic tree with the Maximum Likelihood method in Figure 4 [19]. The strain was closely related to the strains isolated from infected fish (MG063204.1), humans (NR112838.1), and dumevil (MG736237.1).

To further confirm the strain, a series of biochemical experiments were performed using commercial tubes (Hangzhou Binhe Microorganism Reagent Co., Ltd., Hangzhou, China) with three independent experiments (see Appendix A). The results of peptone, Simmons citrate, catalase, lactose, D-mannose, Voges-Proskauer test, glucose (gas), indole, lysine, nitrate reduction, and glycerin tests were positive, and the results were negative for urea, esculin, sucrose, arabinose, xylose, sorbitol, H2S, phenylalanine, and ornithine, which matched the biochemical characteristics of *A. veronii* [20]. According to the phenotypic features, biochemical characteristics, and 16SrRNA gene phylogenetic analysis, the strain YL4077 was identified as *A. veronii*.

The agglutination test is widely used in the diagnosis of bacterial infections, especially those caused by Gram-negative organisms such as *Aeromonas* spp. [21]. A total of 14 and 187 serum samples from symptomatic and asymptomatic sheep were tested by rapid agglutination test (RAT) with antigen prepared from *A. veronii* YL4077. RATs were performed according to previous studies [22]. Surprisingly, the raters of symptomatic and asymptomatic positivity in sheep were 64.28% (95% CI 52.25–76.31%) and 8.02% (95% CI 6.96–9.08%), respectively, suggesting that *A. veronii* was one of the pathogens causing respiratory disease in a sheep farm.

Thirty-two Balb/c mice were divided into four groups, G1–G4, with 8 mice/group. The *A. veronii* YL4077 overnight culture was serially diluted to obtain the concentrations of 5.5 × 10^7^ CFU/mL, 5.5 × 10^6^ CFU/mL, 5.5 × 10^5^ CFU/mL, and 5.5 × 10^4^ CFU/mL for G1–G4, respectively. The mice were injected intraperitoneally with 200 μL and monitored for mortality for 7 days (see Table 1). The 50% lethal dose (LD50) of bacteria was calculated by the modified Kärber method [23]. Finally, the LD50 in Balb/c mice exposed to *A. veronii* YL4077 was determined to be 1.10 × 10^6^ CFU. The anatomical experiments on the dead mice showed obvious congestion in the lungs and liver, and all of the mice developed severe abscesses at the injection site. The bacteria were isolated in the lungs, liver, and blood of the dead mice again. The above data indicate that the strain has a certain pathogenicity.

To further confirm the pathogenicity of *A. veronii* YL4077 in sheep, 1 mL of the *A. veronii* YL4077 overnight culture (1.10 × 10^9^ CFU/mL) was exposed to an 8-month-old lamb through intraperitoneal injection (the sheep was provided by The Animal Experimental Center, Northwest A&F University). The body temperature was increased to 40.5 °C, accompanied by dyspnoea and bloody diarrhea 4 h post-infection, and the sheep died after 19 h. Upon postmortem examination, severe congestion and edema of the lungs and intestinal tract was observed. *A. veronii* YL4077 was again cultured from the blood, lungs, and liver. We concluded that pathogenic *A. veronii* is the cause of respiratory disease and death in sheep.

We then examined the bacterial resistance of *A. veronii* YL4077 by utilizing the disc diffusion assay (the disc was purchased from Hangzhou Binhe Microorganism Reagent Co., Ltd., Hangzhou, China), while Escherichia coli ATCC 25922 was used as a quality control. The results were interpreted following Clinical and Laboratory Standards Institute (CLSI) guidelines. *A. veronii* YL4077 was sensitive to cefotaxime, gentamicin, and enrofloxacin; intermediately sensitive to doxycycline; and resistant to ampicillin, ceftiofur, amoxicillin, kanamycin, neomycin, streptomycin, tetracycline, florfenicol, and tylosin, as shown in Table 2.

We suggest the combination of cefotaxime and gentamicin as an effective treatment based on the results of the antimicrobial susceptibility test. To our delight, the mortality rate of diseased sheep was reduced to 0.

## 3. Discussion

*Aeromonas* spp. was first isolated from pneumonia, wound infections, septicemia, and abortion in horses, cattle, and pigs in 1987 [24]. It is known as a threat to aquaculture systems and human health, causing a high death rate [3,25]. As an important member of *Aeromonas*, *A. veronii* is pathogenic to both aquatic and terrestrial animals [7,8,26]. To date, there has been no documentation of sheep infected with *A. veronii*. Here, we identified an *A. veronii* strain, YL4077, in a sheep that died from severe respiratory illnesses in Shaanxi Province, China.

In this study, the strain was resistant to ampicillin, ceftiofur, amoxicillin, kanamycin, neomycin, streptomycin, tetracycline, florfenicol, and tylosin, consistent with previous findings. It has been shown that almost 96% of Aeromonas isolates from humans and aquatic animals are resistant to the antibiotics ampicillin, ciprofloxacin, chloramphenicol, tetracycline, and cotrimoxazole. [27,28]. From a therapeutic standpoint, multi-drug-resistant *A. veronii* makes treatment options more limited. Pathogenicity results revealed that the strain is pathogenic to both sheep and mice. Thus, *A. veronii* is not only an important threat to the sheep industry, but also poses a potential threat to human health, as the infection may spread to humans via the consumption of undercooked meat contaminated with *A. veronii* [25].

Respiratory diseases are a widespread problem in sheep flocks [29], with presentations varying from acute deaths to chronic ill-thrift and being either sporadic or involving whole groups of animals. This could be attributed to the full roles of other pathogenic viruses such as PPRV, FMDV, MV, PIV3, and ORSV, especially in young sheep. These viruses can, however, form complex infections with other bacterial species (co-infection), such as *Pasteurella* spp., *Mannheimia* spp., and *Mycoplasma* spp., which could come as a secondary infection leading to acute disease outbreaks [30]. However, all the targeted pathogenic viruses and *Mycoplasma* test results were negative, and no other bacteria were isolated in lung samples except *A. veronii*. One possible cause is that the sheep had been treated with ceftiofur sodium for five consecutive days, which inhibited the proliferation of ceftiofur-sensitive bacteria. We deem that ceftiofur-resistant *A. veronii* infection is the leading cause of sheep death. According to the results of the RAT, 64.28% (9/14) and 7.81% (15/187) of the serums from the symptomatic and asymptomatic sheep were positive, respectively. These results further showcase *A. veronii* as the pathogen causing respiratory disease in sheep.

Previous studies revealed that animals exposed to a high level of ammonia experience irritation of mucous membranes and the respiratory tract [31]. Emerging evidence suggests that even low levels of gaseous ammonia have toxic effects on the respiratory tract and cause an imbalance in nasal microbiota in growing pigs and broiler chickens [32,33]. According to the sheep owner, this area experienced a sudden temperature drop in late March, and the sheepfold doors were closed to keep warm, which resulted in ammonia concentrations exceeding 31 ppm. This would cause adverse effects on the respiratory tract of sheep. Studies have shown that the immunity of animals can be reduced under cold stimulation [34]. Cold-induced stress impaired the production of Th1 protective cytokines and can reduce the level of IL-2 in the blood of mice [35,36]. Borsoi et al. [37] indicated that cold, stress-challenged birds have a decreased capacity to overcome the induced infection. Guo H et al. [38] demonstrated that cold stimulation can weaken the immunity of sheep. Furthermore, alveolar macrophages are the first line of defense against inhaled bacteria [39], and overcrowding stress decreases macrophage activity and increases bacterial infections in broiler chickens [40]. Taken together, these stress stimuli may induce *A. veronii* to infect the sheep and cause respiratory disease. However, the source of the *A. veronii* remains unclear. This intensive sheep farm did not have any trade in recent months, thereby excluding a possibility that *A. veronii* was transmitted via the sheep trade. Previous studies have suggested that *A. veronii* is widely distributed in aquatic environments and soil [5,6,7,8]. We speculate that the source of *A. veronii* was the potable water and soil. An important clue in Figure 1D is intestinal mucosal hemorrhage. Therefore, we boldly speculate that the sheep were infected with *A. veronii* through the digestive tract. Our next step will be to clarify the source of the strain and verify this conjecture.

Considered together, we make the following recommendations: First, a major emphasis should be placed on heightened, strict biosecurity rules on sheep farms, including thorough cleaning and disinfection of sheepfolds. Secondly, make sure that every sheep can drink warm, clean, and safe water, especially in the cold season. Thirdly, attention should be paid to ventilation. Finally, and most importantly, the combination of cefotaxime with gentamicin was an effective treatment for *A. veronii* infection. In conclusion, the strain has a certain pathogenicity that may cause serious, even life-threatening diseases in domestic animals under specific conditions. To our knowledge, this is the first case report of pathogenic *A. veronii* isolated from sheep in China. It is necessary to pay attention to the potential threat of *A. veronii* infection to the sheep industry.

## Figures and Tables

**Figure 1 microorganisms-11-00333-f001:**
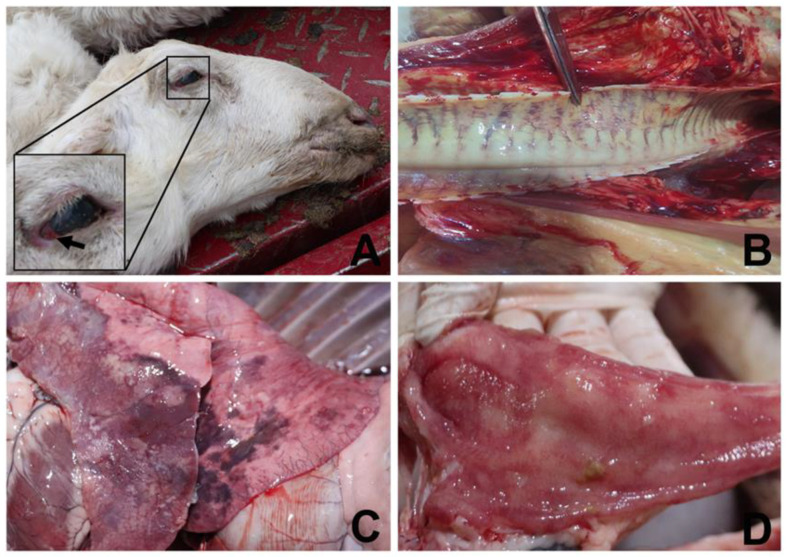
Macroscopic appearance of dead sheep, trachea, lung, and mesentery. (**A**) Congestion of the conjunctiva and nasal discharge were observed in dead sheep; (**B**) Congestive tracheal ring; (**C**) Pulmonary consolidation and congestion; (**D**) Intestinal mucosal hemorrhage.

**Figure 2 microorganisms-11-00333-f002:**
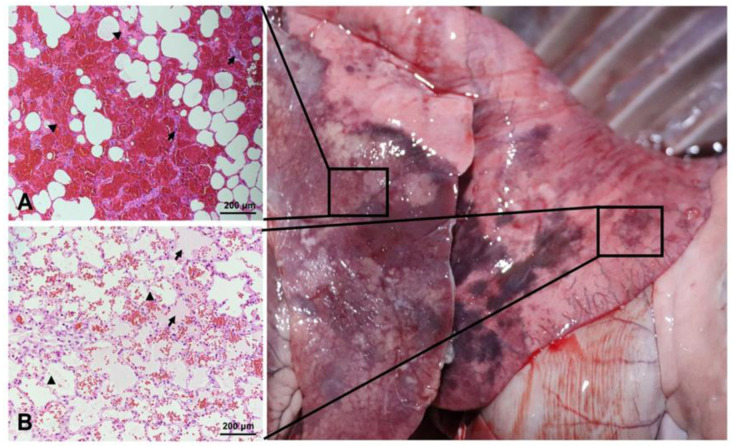
*HE*-stained pathological sections of different parts of the lungs from the sheep. (**A**) Infiltrating lymphocytes (black arrow): the alveolar space of the lung tissue was thickened, and the interstitial lung was congested (black triangle) (bar = 200 µm, 200×); (**B**) fibrinous exudation (black arrow): vacuoles were visible in the alveolar cells (black triangle) (bar = 200 µm, 200×).

**Figure 3 microorganisms-11-00333-f003:**
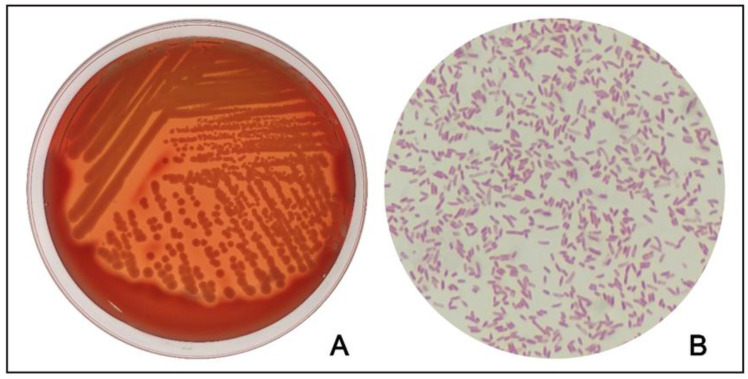
This bacterium is rod-shaped and Gram-negative and shows β-hemolytic activity on ovine blood agar plates. (**A**) This bacterium was incubated on the ovine blood agar plates overnight at 37 °C; (**B**) gram staining showed the bacterium is rod-shaped and Gram-negative.

**Figure 4 microorganisms-11-00333-f004:**
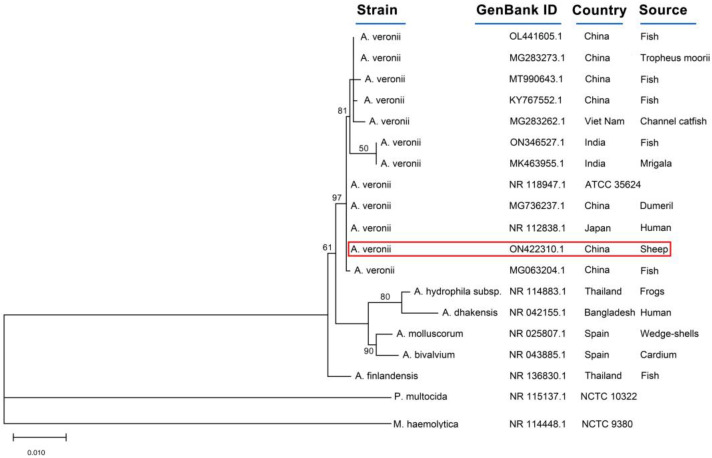
Phylogenetic tree of 16SrRNA in *A. veronii* YL4077 isolated from dead sheep in China. A red line indicates the isolate from this study. The numbers in each node are the confidence values. The distance bar is the number of nucleotide substitutions per site.

**Table 1 microorganisms-11-00333-t001:** Experimental data on the pathogenicity of *A. veronii* YL4077 in mice.

Group	Dose		Numbers (n)	Death Number (n)	Mortality (P)	P2
	CFU/Mouse	Log				
G1	1.1 × 10^7^	7.04	8	8	1	1
G2	1.1 × 10^6^	6.04	8	6	0.75	0.5625
G3	1.1 × 10^5^	5.04	8	2	0.25	0.0625
G4	1.1 × 10^4^	4.04	8	0	0	0
lgLD50 = X − d(∑Pi − 0.5); Standard error SlogLD50 = d√((∑P − ∑P2)/(n − 1))					∑P = 2.00	∑P^2^ = 1.625

“X” means the maximum logarithmic dose, “d” means the difference in logarithmic dose between two adjacent groups, “Pi” means the mortality rate, and “i” means the group number.

**Table 2 microorganisms-11-00333-t002:** Results of the drug susceptibility test.

Drugs	Diameter (mm)	Sensitivity	Judging Criteria (mm)			Dose (ug/disc)
			R	I	S	
Amp	0	R	≤18	19–21	≥22	10
Cef	13	R	≤14	15–22	≥23	30
Cet	17	S	≤14	15–22	≥23	30
Amo	0	R	≤13	14–17	≥18	10
Kan	12	R	≤13	14–17	≥18	30
Gen	19	S	≤14	-	>14	10
Neo	10	R	≤12	13–16	≥17	30
Str	9	R	≤11	12–14	≥15	10
Tet	13	R	≤14	15–18	≥19	30
Dox	13	I	≤12	13–15	≥16	30
Flo	12	R	≤12	13–17	≥18	30
Enr	23	S	≤12	13–16	≥17	10
Tyl	13	R	≤14	14–15	≥16	30

Note: Amp Ampicillin; Cef, Ceftiofur; Cet, Cefotaxime; Amo, Amoxicillin; Kan, Kanamycin; Gen, Gentamicin; Neo, Neomycin; Str, Streptomycin; Tet, Tetracycline; Dox, Doxycycline; Flo, Florfenicol; Enr, Enrofloxacin; Tyl, Tylosin; R, Resistance; I, Intermediate; S Sensitive.

## Data Availability

The authors confirm that all data are fully available without restriction.

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
