# Peer review of "Isolation and Identification of Aeromonas veronii in Sheep with Fatal Infection in China: A Case Report"

_microorganisms, 2023, doi:10.3390/microorganisms11020333_

Round 1

Reviewer 1 Report

Isolation and identification of Aeromonas veronii in sheep with 2 fatal infection in China: a case report represents an interesting report and a very epidemiological study. 

Author Response

Dear Reviewer

Thank you very much for your recognition of our work. We highly appreciate the time and efforts you have paid into this manuscript

If there are other errors or further revision requests, please notify us by e-mail.

Yours Sincerely,

Yongqiang, Miao,

College of Veterinary Medicine,

Northwest A&F University,

Yangling, Shaanxi,

People's Republic of China, 712100.

E-mail, 451233165@qq.com

Reviewer 2 Report

The work is of very good quality and deserves to be considered, however, there are some points to complete or improve so that the work becomes fully understandable, lucid and especially reproducible. 

Point 1: in the introduction (lines 29 to 41): The introduction lacks information on the sheep population in the province or country as well as trade figures showing the importance of the health risk probably caused by respiratory infections.  

Point 2: Line 43: Give the exact position of the site 

Point 3: lines 46 - 48: When are the animals vaccinated? Is there a link between the inflammatory reaction of the vaccine and the appearance of symptoms in the farm?   

Point 4: Line 50: For how long? If it is possible? 

Point 5: Line 52: give the age please.

Point 6: Lines 59-60: Please, give the complete protocol and materials with references. 

point 7: Line 67 : reference of the microscope and protocol please:

Point 8: Line 70 : Why only the lung ? blood , spleen or mesenteric lymph nodes would also be good indicators of infections.

Point 8: Line 75 : Why only the lung ?

Point 9: Lines 70-73: how about Salmonella or Pasteurella ? 

Point 10: Lines 77 - 78: Have you tried other conditions to isolate fastidious bacteria?   

Point 11: Line 83: It would not be appropriate to confirm so soon .

Point 12: Line 92 -93: Reference and PCR  protocol  

Point 13: Line 93: the obtained sequences. 

Point 14: Line 103: Insignificant tree. It is necessary to change the outgroup and to put other respiratory pathogenic species and gamma-proteobacteria. 

Point 15: Line 114: i stay septic to this (connfirmation)  https://doi.org/10.1128/JCM.01963-14

Point 16: Line 120: Would it not be linked to asymptomatic carriage? 

Point 17: Line 161: You need to talk about the probable sources of contamination. how is this farm contaminated? do you have any leads? (without mentioning the conditions in which the symptoms appears)

Point 18: Lines 169-170: see the point 1. 

Point 19: Lines 171-172: Give the reference please 

Point 20: Line 201-202: Why focus only on the respiratory origin? the digestive starting point is also important. it should at least be mentioned! (To make things simple: Your mice and sheep were infected by the general route and the diffusion was done by blood and lymphatic routes!)

Author Response

Dear Reviewer

Thank you for kindly reviewing the manuscript entitled “Isolation and identification of Aeromonas veronii in sheep with fatal infection in China: a case report” (Manuscript ID: microorganisms-2126351). We thank you for your professional evaluation and suggestions and the time spent analyzing this manuscript. We have made corresponding explanations, supplements, and corrections in the following modules according to your comments. Following is our point-by-point response to these concerns. Main revisions and track changes are highlighted in red color for review only.

Point 1: in the introduction (lines 29 to 41): The introduction lacks information on the sheep population in the province or country as well as trade figures showing the importance of the health risk probably caused by respiratory infections.

Response1:

We would like to thank you for your positive comments and valuable suggestions to improve our manuscript. We have revised these mistakes in the manuscript, and the detailed corrections are listed below.

【Manuscript Line 41-42】

In this report, A. veronii was first isolated in a sheep that died from respiratory disease. This intensive sheep farm didn’t have any sheep trade in recent months.

Point 2: Line 43: Give the exact position of the site

Response 2:

We feel great thanks for your professional review work on our article. According to your comments, we have supplemented the exact position of the site. Once again, we acknowledge your comments very much, which are valuable in improving the quality of our manuscript. and the detailed corrections are listed below.

【Manuscript Line 45-46】

One intensive commercial Hu sheep farm in north Shaanxi Province, China, with an altitude range of 1266 m, a longitude range of 109.02°, and a latitude range of 38.20°

Point 3: lines 46 - 48: When are the animals vaccinated? Is there a link between the inflammatory reaction of the vaccine and the appearance of symptoms in the farm?

Response 3:

We feel great thanks for your valuable comments. The herd had been vaccinated with Foot and Mouth Disease Vaccine (Xinjiang TianKang Bio-technique Co. Ltd., strain OHM/02 and AKT-III, inactivated) and Peste des Petits Ruminants Vaccine (Xinjiang TianKang Bio-technique Co. Ltd. strain Clone9, inactivated) on March 6 and March 15, respectively. The above vaccines are the most widely used vaccines in China in recent years. It is known to be safe and reliable. The inflammatory reaction of the vaccine was low after two weeks post-vaccination. Therefore, we speculate that there was little correlation between the inflammatory reaction of the vaccine and the appearance of symptoms on the farm.

Point 4: Line 50: For how long? If it is possible?

Response 4:

Thanks for your valuable comments and for pointing out this omission. According to your comments, we have supplemented the exact position of the site.

【Manuscript Line 53-54】

Multi-drug treatment strategies (such as penicillin 20,000U/Kg, ceftiofur 5 mg/kg, and tylosin 10 mg/kg, repeated twice per day for 7 days.) have been attempted to improve clinical outcomes, but with limited success.

Point 5: Line 52: give the age please.

Response 5:

Thank you for your professional review of our article. We have added the detailed information, and the detailed corrections are listed below.

【Manuscript Line 57-58】

On April 15, 2022, we went to the farm and happened to have a sheep (a 15-month-old mature ewe) die from respiratory disease.

Point 6: Lines 59-60: Please, give the complete protocol and materials with references.

Response 6:

Thank you for your professional review of our article. We have added the detailed information, and the detailed corrections are listed below.

【Manuscript Line 63-65】

In addition, 14 and 187 blood samples were collected and centrifuged (5000 g, 10 min) to gather serum in symptomatic and asymptomatic sheep.

Reference

[12] Shi SY, Lu SY, Si T, et al. DJ-1 links muscle ROS production with metabolic reprogramming and systemic energy homeostasis in mice. Nat Commun. 2015,6.

point 7: Line 67: reference of the microscope and protocol please:

Response 7:

Thank you very much for your comments. We have added the microscope and protocol.

the detailed corrections are listed below

【Manuscript Line 72-73】

The sections were observed with an Olympus BX51 microscope (Olympus Corporation) [13]

Reference

[13] Calvi LM, Bromberg O, Rhee Y, et al. Osteoblastic expansion induced by parathyroid hormone receptor signaling in murine osteocytes is not sufficient to increase hematopoietic stem cells. Blood. 2012;119(11):2489-2499.

Point 8: Line 70: Why only the lung? blood, spleen, or mesenteric lymph nodes would also be good indicators of infections. Line 75: Why only the lung?

Response 8:

We are very grateful for your comments and thoughtful suggestions.

Line 70 and Line 75: According to the analysis of the clinical symptoms and anatomical observations, lung involvement is the leading cause of sheep death. We focused on the pathogens and pathological changes in the lung. In addition, A. veronii can be isolated from the blood of dead sheep.

Point 9: Lines 70-73: how about Salmonella or Pasteurella?

Response 9:

Thank you for your professional review of our article. We have tried our best to isolate the potential pathogenic microbes. However, no other bacteria were isolated in lung samples but A. veronii. One possible cause is the sheep had been treated with ceftiofur sodium for five consecutive days, which inhibited the proliferation of ceftiofur-sensitive bacteria.

Point 10: Lines 77 - 78: Have you tried other conditions to isolate fastidious bacteria?

Response 10:

We are very grateful for your comments. We have tried a nutrient agar containing 5% sheep blood and inoculation under anaerobic and aerobic at 37°C for 24 h. There are several potential causes. First of all, A. veronii inhibits the growth of other bacteria. Secondly, the previous antibiotic treatment affects the isolation results of fastidious bacteria.

Point 11: Line 83: It would not be appropriate to confirm so soon.

Response 11:

We are very grateful for your comments and thoughtful suggestions. We have revised these mistakes in the manuscript. The detailed revision was shown as follows

【Manuscript Line 89】

Figure 2. HE-stained pathological sections of the different parts of lungs from the sheep.

Point 12: Line 92 -93: Reference and PCR protocol 

Response 12:

We are very grateful for your comments and thoughtful suggestions. We have revised

the mistake in the manuscript. The detailed revision was shown as follows.

【Manuscript Line 98-100】

The PCR conditions consisted of 94 °C for 5 min, followed by 35 cycles of 94 °C for 30 s, 55 °C for 45 s, and 72 °C for 90 s, and a final extension at 72 °C for 6 min.

Point 13: Line 93: the obtained sequences.

Response 13:

Thanks for your careful checks. We feel sorry for the improper wording. According to your suggestion, we have corrected the manuscript. The detailed corrections are listed below.

【Manuscript Line 100】

The obtained sequences were sequenced at Beijing Tsingke Biotechnology Co., Ltd.

Point 14: Line 103: Insignificant tree. It is necessary to change the outgroup and to put other respiratory pathogenic species and gamma-proteobacteria.

Response 14:

Thank you for your constructive comment. We absolutely agreed with your opinion.

Here, the phylogenetic tree elucidates the relationships among A. veronii isolated in sheep with Aeromonas spp. According to your opinion, we have changed the outgroup and put other respiratory pathogenic bacteria such as Pasteurella multocida, and Mannheimia haemolytica. The detailed correction is in line 108

Point 15: Line 114: i stay septic to this (connfirmation) https://doi.org/10.1128/JCM.01963-14

Response 15:

Thank you for your constructive comment.

In this study (https://doi.org/10.1128/JCM.01963-14), the strain collection used was composed of 51 Aeromonas spp. collected from diarrheagenic patients. The study indicated that the Aeromonas spp may be an important foodborne pathogen.

In our report, A. veronii can be isolated from the blood which demonstrated the sheep is septic. In addition, all of the mice developed severe abscesses at the injection of A. veronii (Line 137).

Point 16: Line 120: Would it not be linked to the asymptomatic carriage?

Response 16: Thanks for your comments.

In this report, the antigen-antibody reaction is positive indicating the asymptomatic sheep were also infected with A. veronii. The asymptomatic carriage cannot be demonstrated.

Point 17: Line 161: You need to talk about the probable sources of contamination. how is this farm contaminated? do you have any leads? (Without mentioning the conditions in which the symptoms appear)

Response 17:

We thank you very much for your comments and thoughtful suggestions. We absolutely agreed with your opinion. we feel that the appropriate place for these was in line 213. The detailed corrections are listed below.

【Manuscript Line 217-223】

However, the source of A. veronii remains unclear. This intensive sheep farm didn’t have any trade in recent months, thereby excluding the possibility that A. veronii transmission via the sheep trade. Previous studies have suggested that A. veronii is widely distributed in aquatic environments and soil [5, 6, 7, 8]. We speculate that the source of A. veronii was potable water and soil. Our next step was to clarify the source of the strain.

Point 18: Lines 169-170: see the point 1.

Response 18:

We thank you very much for your comments and thoughtful suggestions. We absolutely agreed with your opinion. we feel that the appropriate place for these was in line 213-214. The detailed corrections are listed below.

【Manuscript Line 217-218】

This intensive sheep farm didn’t have any trade in recent months, thereby excluding the possibility that A. veronii transmission via the sheep trade.

Point 19: Lines 171-172: Give the reference please

Response 19:

We thank you very much for your comments and thoughtful suggestions. In this report,

pathogenicity results revealed that the strain is pathogenic to both sheep and mice. We therefore speculate that A. veronii poses a potential threat to human health for the infection may spread to humans via the consumption of undercooked meat contaminated with A. veronii. We have added the needed references, and the detailed revision was shown as follows.

【Manuscript Line 182-185】

Thus, A. veronii is not only an important threat to the sheep industry but also poses a po-tential threat to human health for the infection may spread to humans via the consump-tion of undercooked meat contaminated with A. veronii [23].

[23].       Pessoa, R.B.; Oliveira, W.F.; Correia, M.; Fontes, A.; Coelho, L.; Aeromonas and Human Health Disorders: Clinical Ap-proaches. Front. Microbiol. 2022, 13, 868-890

Point 20: Line 201-202: Why focus only on the respiratory origin? the digestive starting point is also important. it should at least be mentioned! (To make things simple: Your mice and sheep were infected by the general route and the diffusion was done by blood and lymphatic routes!)

Response 20:

We thank you very much for your comments and thoughtful suggestions. We absolutely agreed with your opinion. we feel that the appropriate place for these was in line 217-219. The detailed corrections are listed below.

【Manuscript Line 221-223】

An important clue in Figure 1D is intestinal mucosal hemorrhage. Therefore, we boldly speculate that the sheep infected A. veronii through the digestive tract.

Thank you again for your positive comments and valuable suggestions to improve the

quality of our manuscript.

If there are other errors or further revision requests, please notify us by e-mail.

Yours Sincerely,

Yongqiang, Miao,

College of Veterinary Medicine,

Northwest A&F University,

Yangling, Shaanxi,

People's Republic of China, 712100.

E-mail, 451233165@qq.com

Reviewer 3 Report

Please,

See attached file.

Author Response

Response to Reviewer 3 Comments

Dear Reviewer

Thank you for kindly reviewing the manuscript entitled “Isolation and identification of Aeromonas veronii in sheep with fatal infection in China: a case report” (Manuscript ID: microorganisms-2126351). We thank you for your professional evaluation and suggestions and the time spent analyzing this manuscript. We have made corresponding explanations, supplements, and corrections in the following modules according to your comments. Following is our point-by-point response to these concerns. Main revisions and track changes are highlighted in the yellow color background for review only.

Abstract: Authors should include information about mortality rate and ages (and physiological status) of the animals.

Response:

We thank you very much for your comments and thoughtful suggestions. We have revised these mistakes in the manuscript and the detailed corrections are listed below.

【Manuscript Line 11-15】

According to the findings of a sheep breeding farm in Shaanxi, China, 2.53% (15/594) of sheep exhibited respiratory (clinical) symptoms such as dyspnoea, nasal discharge, wet cough, fever, and progressive emaciation. Although multi-drug treatment strategies (including ampicillin, tylosin, florfenicol, and ceftiofur) have been attempted to improve clinical outcomes, they only met with limited success, and with a mortality rate of 40%.

Line 12: I suggest changing runny nose by nasal discharge.

Response:

We thank you very much for your comments and thoughtful suggestions. We have taken on board your comment and changed runny nose by nasal discharge in Line 12.

Line 14: et al??? What do you mean??? Include all the drugs used and delete “et al”

Response:

We a very grateful to the reviewer for this comment. We have corrected these mistakes based on your suggestions. The corresponding revisions were marked in line 14.

Line 16: sheep positive to A. veronii

Response:

We thank you very much for your comments and thoughtful suggestions. We have corrected these mistakes based on your suggestions. The corresponding revisions were marked in line 17

Line 18: what testing?

Response:

We thank you very much for your comments and for pointing out this omission. We have revised these mistakes in the manuscript and added needed references and the detailed corrections are listed below.

【Manuscript Line 18-19】

The results of the antibiotic susceptibility tests revealed that the strain was sensitive to cefotaxime…

Line 17 and 18: Delete or rewrite this sentence. “Moreover...”

Response: Thank you for this helpful suggestion. The rewritten sentence is:

【Manuscript Line 18】

Pathogenicity tests demonstrated that A. veronii is pathogenic to sheep and mice.

Line 36: China and elsewhere

We feel great thanks for your professional review work on our article.

Here, we want to highlight that it was the first report of A. veronii causing disease in sheep in China.

Line 39 and elsewhere in the manuscript: NEVER use “etc.” Delete and complete the information.

Response:

We thank you for this comment which is absolutely correct. The rewritten sentence is:

【Manuscript Line 40】

Respiratory diseases lead to high economic losses in the sheep industry worldwide. Several pathogens cause respiratory disease, such as Pasteurella multocida, Mycoplasmosis, Mannheimia haemolytica, and Staphylococcus aureus [9, 10, 11].

Line 39: in how many dead sheep?

Response:

We thank you very much for your comments. Likewise, we feel sorry that we did not provide enough information about this. The rewritten sentence is: In this report, A. veronii was first isolated in a sheep that died from respiratory disease.

Line 43: In this paragraph, authors need to complete the information. This is an international journal with readers worldwide that might not know Chinese breeds (add reference to the breed) and the farming system. Please explain.

Response:

Thank you for your professional and detailed suggestion. We have further completed the information on the farm and location. The farming system is intensive commercial farm. Chinese Hu sheep (Ovis aries) is one of the most important sheep breeds in China.

The detailed corrections are listed below.

【Manuscript Line 45-46】

One intensive commercial Hu sheep farm in north Shaanxi Province, China, with an altitude range of 1266 m, a longitude range of 109.02°, and a latitude range of 38.20°, reported that about 2.53% (15/594) of the sheep experienced respiratory

Line 45: 41.2 is your reference number for hypertermia? the average of the diseased? Please rewrite and clarify.

Response:

Thank you for your professional and detailed suggestion. Rectal temperature was maintained at the normal body temperature for sheep (38.5 ℃). 41.2℃ is the average body temperature of the diseased. The detailed corrections are listed below.

【Manuscript Line 48-49】

symptoms such as dyspnoea, runny nose, wet cough, and hyperthermia and hyperthermia (41.2℃, the average body temperature of the diseased).

Line 45: This case reprot need to provide the exact mortality rate. Furthermore, I STRONGLY SUGGEST including a descriptive table with all the sick and dead animals affected by this outbreak. Info about sex, age, clinical signs, treatment, submitted to lab analysis or not, and destiny.

Response:

Thank you for your professional and strong suggestion. We absolutely agreed with your opinion. According to your advice, we added a descriptive table and listed below.

Supplementary Table1 Description of sick and dead sheep affected by this outbreak

sick

dead

Rate

2.53% (15/594)

40.0% (6/15)

Gender

Male

6/15

1/6

Female

9/15

5/6

Age (month)

<6

8/15

3/6

6-12

5/15

1/6

>12

2/15

2/6

Clinical signs

dyspnoea, nasal discharge, wet cough, hyperthermia

dyspnoea, nasal discharge, diarrhea, wet cough, emaciation

Treatment

ampicillin, tylosin, florfenicol, ceftiofur

Lab analysis (YES or NO)

YES

YES

Lines 56-60: Were these sheep treated or not? Detail.

Response:

Thank you for your comment.

The sheep had been treated with ceftiofur sodium (5 mg/kg body weight, intramuscular, Qilu Animal Health Products Co., Ltd) for five consecutive days. Shown in line 58-59 and marked in yellow.

Line 139: Why intra peritoneal? This needs an explanation and discussion

Response:

Thank you for your comment.

Intra-peritoneal injection is one of the most frequently used methods to evaluation of bacteria pathogenicity. Reference to

Lu XJ, Chen J, Yu CH, et al. LECT2 protects mice against bacterial sepsis by activating macrophages via the CD209a receptor. J Exp Med. 2013;210(1):5-13. doi:10.1084/jem.20121466.

Chin W, Zhong G, Pu Q, et al. A macromolecular approach to eradicate multidrug resistant bacterial infections while mitigating drug resistance onset. Nat Commun. 2018;9(1):917. Published 2018 Mar 2. doi:10.1038/s41467-018-03325-6.

Because of infection route of the A. veronii could not be determined. Therefore, we used the intraperitoneal route to the evaluation of A. veronii pathogenicity.

Line 155: After all..How did this case ended??? What was the mortality rate? And Respiratory incidence? 0??? Please add a paragraph explaining.

Response:

We are very grateful for your comments and thoughtful suggestions. We have revised

these mistakes in the manuscript. The detailed revision was shown as follows.

【Manuscript Line 162-164】

We suggest the combination of cefotaxime and gentamicin as an effective treatment according to the results of the antimicrobial susceptibility test. To our delight, the mortality rate of diseased sheep was reduced to 0.

Line 167: etc??

Response:

Thank you for your professional and detailed suggestion. We have deleted ‘etc’ and completed the information in line 180

Line 193: PPM? ppm?

Response:

Thanks for your careful checks. We feel sorry for the improper units. According to

your suggestion, we have corrected the “PPM” into “ppm” in the manuscript. The detailed corrections shown in line 207.

Line 202: Does it mean that A. veronii is ubicuous of the Upper Respiratory Tract? or it is transmitted via aerogena? or both? Please, explain.

Response:

We feel great thanks for your professional review work on our article. Likewise, we

feel sorry that we did not provide accurate information about this. it doesn’t mean that A. veronii is ubicuous of the Upper Respiratory Tract or transmitted via aerogene. A. veronii is ubiquitously distributed in diverse environments. The source of infection for the sheep could not be determined specifically. As a next step, we plan to research the source and infection route.

We have revised the sentence. Detailed revision was shown as follows.

【Manuscript Line 215-216】

Taken together, these stress stimuli may induce A. veronii to infect the sheep and cause respiratory disease.

Line 208: How was this treatment in the herd? Prophylactic? metaphilaxis? Individualized (how was the decision taken to treat or not?). Please, clarify.

Response:

We are very grateful for your comments and thoughtful suggestions. The antibiotic therapy in the herd was targeting the individual processes. (Finally, and most importantly, the combination of cefotaxime with gentamicin was an effective treatment for A. veronii infection.)

The prophylactic measures in the herd are in following: At first, a major emphasis was placed on heightened strict biosecurity rules on sheep farms, thorough cleaning, and disinfection of sheepfolds. Secondly, make sure that every sheep can drink warm, clean, and safe water, especially in the cold season. Thirdly, attention should be paid to ventilation.

Thank you again for your positive comments and valuable suggestions to improve the

quality of our manuscript.

If there are other errors or further revision requests, please notify us by e-mail.

Yours Sincerely,

Yongqiang Miao,

College of Veterinary Medicine,

Northwest A&F University,

Yangling, Shaanxi,

People's Republic of China, 712100.

E-mail, 451233165@qq.com

Round 2

Reviewer 3 Report

-